

# Recombinant expression in *E. coli* of human FGFR2 with its transmembrane and extracellular domains

Adam Bajinting[1,2] and Ho Leung Ng[1,3]

[1] Department of Chemistry, University of Hawaii at Manoa, Honolulu, HI, United States of America
[2] School of Medicine, St. Louis University, St. Louis, MO, United States of America
[3] University of Hawaii Cancer Center, Honolulu, HI, United States of America

## ABSTRACT

Fibroblast growth factor receptors (FGFRs) are a family of receptor tyrosine kinases containing three domains: an extracellular receptor domain, a single transmembrane helix, and an intracellular tyrosine kinase domain. FGFRs are activated by fibroblast growth factors (FGFs) as part of complex signal transduction cascades regulating angiogenesis, skeletal formation, cell differentiation, proliferation, cell survival, and cancer. We have developed the first recombinant expression system in *E. coli* to produce a construct of human FGFR2 containing its transmembrane and extracellular receptor domains. We demonstrate that the expressed construct is functional in binding heparin and dimerizing. Size exclusion chromatography demonstrates that the purified FGFR2 does not form a complex with FGF1 or adopts an inactive dimer conformation. Progress towards the successful recombinant production of intact FGFRs will facilitate further biochemical experiments and structure determination that will provide insight into how extracellular FGF binding activates intracellular kinase activity.

## INTRODUCTION

As receptor tyrosine kinases (RTKs), Fibroblast growth factor receptors (FGFRs) have three primary domains: an extracellular domain (ECD), a single transmembrane helix (TM), and an intracellular tyrosine kinase domain (KD) (Fig. 1). These proteins are expressed primarily in endothelial, fibroblast, vascular smooth muscle, neuroectodermal, and mesenchymal cells. When activated by fibroblast growth factors (FGFs), these receptors are responsible for activating mechanisms via trans-autophosphorylation that result in angiogenesis, skeletal formation, and cell differentiation, proliferation, survival, and growth. Within the subfamily are four types of FGFRs: FGFR1, FGFR2, FGFR3, and FGFR4, which share 55–72% sequence homology. Due to their critical roles in cell and tissue development, mutations of FGFRs are known to lead to achondroplasia (poor cartilage growth) and developmental disorders that exhibit craniosynostosis (improper skull formation) (*Turner & Grose, 2010*). FGFR2 and FGFR3 have also been implicated in cancers such as bladder

Corresponding author
Ho Leung Ng, hng@hawaii.edu

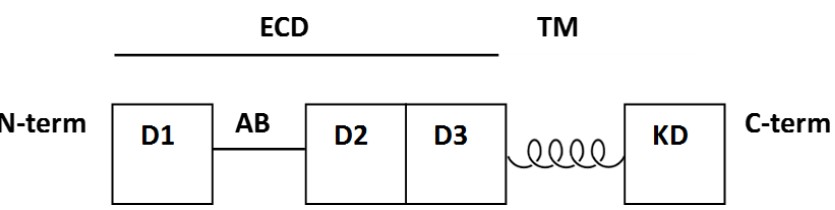

**Figure 1  Schematic of domains and motifs in FGFR2.** D1–D3 are the immunoglobulin domains. AB is the acid box motif. ECD is the extracellular domain (or ectodomain). TM is the single transmembrane helix. KD is the intracellular kinase domain.

cancer, and inhibitors are being investigated as potential cancer therapeutics (*Turner & Grose, 2010*; *Brooks, Kilgour & Smith, 2012*; *Daniele et al., 2012*; *Dieci et al., 2013*).

Crystal structures have been determined of the ectodomains and kinase domains of the FGFRs (*Mohammadi, Schlessinger & Hubbard, 1996*; *Plotnikov et al., 1999*; *Plotnikov et al., 2000*; *Schlessinger et al., 2000*; *Yeh et al., 2002*; *Zhang et al., 2009*). The ectodomain is composed of three immunoglobulin (Ig) domains termed D1, D2, and D3. Between D1 and D2 is an acid box motif, a sequence of 20 acid-rich amino acids that binds to divalent cations to stabilize the interaction between FGFR and heparin/heparin sulfate proteoglycans (HSPGs) (*Patstone & Maher, 1996*). The acid box also mediates interactions with other proteins (*Sanchez-Heras et al., 2006*) and plays a key role in auto-inhibition (*Kalinina et al., 2012*). For the FGFR2 ECD + TM construct in particular, the structure of the ECD lacks both the acid box and the D3 domain. Removal of both regions increases the affinity for heparin and the ability of FGF to active FGFR (*Wang et al., 1995*). The D3 domain is unnecessary for FGF1 activation and is involved in differential responses to different FGFs (*Yu et al., 2000*).

There are many open questions about the structure of FGFRs regarding the transmembrane helix and how it connects the ECD and KD. There is an NMR structure of the FGFR3 TM that shows it as a single alpha helix (*Bocharov et al., 2013*). However, the biological relevance of this structure is unclear as the data was collected from a construct containing only the TM and the extracellular juxtamembrane region, without the ECD or KD. As the TM represents a tiny proportion of the full-length FGFR, it is likely that the natural conformation of the TM in the intact receptor *in vivo* differs significantly from the isolated peptide.

X-ray crystallography of a multi-domain construct containing the TM would provide more insight into the receptor activation mechanism and how activation status is transduced across the membrane. *Bocharov et al. (2013)* proposed a "string puppet theory" mechanism of signal transduction based on the NMR structure of the TM helix. The string puppet theory proposes that FGFR dimerizes in an inactive form via its transmembrane domains without FGF and heparin; the active conformation results when the inactive dimer binds to FGFs. Details of the stoichiometry of FGF, heparin, and FGFR in the activated complex are also debated (*Lemmon & Schlessinger, 2010*). Conclusively resolving different hypotheses about inactive and active FGFR states will come from detailed structures of intact FGFR.
**Table 1  FGFR2 and FGFR3 constructs created.**

| Construct | Expected size (kDa) |
| --- | --- |
| MBP + FGFR2 31-406 (ECD + TM) | 71.5 |
| MBP + FGFR2 370-651 (TM + KD) | 73.7 |
| MBP + FGFR2 31-651 (ECD + TM + KD) | 111.5 |
| MBP + FGFR3 143-405 (ECD + TM) | 71.3 |
| MBP + FGFR3 365-771 (TM + KD) | 87.9 |
| MBP + FGFR3 143-771 (ECD + TM + KD) | 112.3 |

Here we describe our development of a recombinant expression system in *E. coli* to produce significant quantities of functional FGFR with its TM linked to either its ECD or KD for eventual structural studies. Recombinant expression of complex eukaryotic proteins in *E. coli* is often challenging and results in low yields of insoluble, inactive protein (*Rosano & Ceccarelli, 2014*). Expression of membrane proteins containing the very hydrophobic transmembrane domains is especially problematic (*Hattab et al., 2015*). Moreover, there have been only a few studies describing the successful heterologous expression of protein kinases including their transmembrane domains, with none expressed in *E. coli* (*Mi et al., 2008*; *Mi et al., 2011*; *Lu et al., 2012*; *Paavilainen et al., 2013*; *Opatowsky et al., 2014*; *Chen, Unger & He, 2015*). These prior studies describe the recombinant expression of EGFR, EphA2, PDGFR, and Kit. Here, we describe the expression of constructs of FGFR2 and FGFR3 containing ECD + TM in *E. coli* in sufficient yield for protein crystallization. FGFR was expressed as a fusion protein with maltose binding protein (MBP), which has been shown to improve expression yield and solubility (*Kapust & Waugh, 1999*). We show that the FGFR2 ECD + TM construct is functional in binding heparin and dimerizing. Our simple recombinant method will facilitate biochemical experiments studying the relationship between the TM and other domains.

## MATERIALS & METHODS

### DNA cloning of constructs

Polymerase incomplete primer extension (PIPE) cloning was used to obtain specific domain combinations of FGFR2, and the cloning vector pSpeedET with an N-terminal *E. coli* maltose binding protein (MBP) fusion tag of 42.5 kDa (*Klock & Lesley, 2009*). The domain combinations created are shown in Table 1. The FGFR inserts were amplified by PCR using Phusion Hi Fidelity DNA Polymerase, 200 mM dNTP, 0.5 μM forward and reverse primers, and 6% DMSO. PCR products were extracted from agarose gel and purified using Thermo Scientific GeneJet Gel Extraction Kits. The MBP fusion tag was added to the construct to improve construct solubility and expression (*Kapust & Waugh, 1999*), allow purification by amylose affinity chromatography, and identification by Western blot with an anti-MBP antibody (New England Biolabs (E-8038)). Cloning results were confirmed by DNA sequencing.

## Small-scale expression

Small scale expression studies were performed using *E. coli* Lemo21 cells (New England Biolabs). 10 mL inoculate from an overnight culture was added to 100 mL of TB media and shaken at 37 °C. The $OD_{600}$ was monitored as it approached an absorbance of 0.6. Once the culture reached an $OD_{600}$ of 0.4–0.5, the cells were cooled to 18 °C in the shaker to slow the growth of cells and rhamnose was added to a final concentration of 0.5 mM to titrate expression levels in the Lemo21 cells. Once growth reached OD of 0.6, 1 mL of each construct culture was taken to serve as a negative control for later experiments. Isopropyl β-D-1-thiogalactopyranoside (IPTG) was then added to 0.1 mM final concentration to each culture to induce expression. The cells were then grown in a shaker at 18 °C overnight.

## Harvesting and lysing cells

Each of the cultures was centrifuged at 4 °C at 4,800 g for 10 min. The culture media was discarded, and the pellet was washed by resuspending in lysis buffer (300 mM NaCl, 50 mM HEPES at pH 7.5, 0.1 mM $MgSO_4$, 5% glycerol, 0.5 mM TCEP, benzamidine, and PMSF). It was centrifuged at 4,800 g for 10 min, after which the lysis buffer was discarded. 20 mg of post induction *E. coli* cell pellet was resuspended in 180 μL of lysis buffer (300 mM NaCl, 50 mM HEPES at pH 7.5, 0.1 mM $MgSO_4$, 5% glycerol, 0.5 mM TCEP, benzamidine, and PMSF). 20 μL of 10 mg/mL lysozyme stock was added in addition to 0.3 μL of DNAse I. Next, the lysis reaction was put through three freeze-thaw cycles to lyse the cells.

## Western blot analysis

Western blotting was performed on PVDF membranes after wet transfer from polyacrylamide gels. Membranes were blocked with Amresco RapidBlock solution for 5 min and then incubated with HRP-conjugated anti-MBP monoclonal antibody (New England Biolabs) overnight at 4 °C. Membranes were then washed three times for 5 min with 20 mM Tris–HCl pH 7.5, 150 mM NaCl, and 0.1% Tween 20. Finally, the blots were developed using the KPL TMB Membrane Peroxidase Substrate System kit.

## Large-scale expression studies

After the best candidates for continued expression studies were determined, the FGFR2 ECD + TM constructs were expressed at a larger scale. The expression procedures (transformation and inoculation) are identical except that instead of 10 mL of initial culture (in LB) to inoculate 100 mL of TB, 100 mL of initial culture was grown and inoculated into 1,000 mL of TB.

Once the culture reached $OD_{600}$ of 0.4–0.5, the cells were cooled to 18 °C in the shaker to slow growth, and rhamnose was added to a final concentration of 0.5 mM to titrate expression levels in the Lemo21 cells. Isopropyl β-D-1-thiogalactopyranoside (IPTG) was then added at 0.1 mM final concentration to induce expression.

## Cell lysis

Each construct's cell pellet was resuspended in lysis buffer by vortexing and physically mixing with a pipet to ensure homogeneity. 1 μL of DNAse I was added in addition to 1 μM final concentration of $CaCl_2$, and additional protease inhibitors (E-64, pepstatin, and

bestatin) prior to lysis by sonication at 4 °C with a Fisher Scientific P-550. Sonication was performed for a total of 2 min, divided into 20 s of sonication followed by 40 s of rest (total of 6 min of clock time), at 60% of full power. Samples are kept on ice during sonication. After sonication, the suspensions were centrifuged at 48,400 g for 30 min.

## Detergent extraction of FGFR from cell membranes

Unlike the small-scale expression trials, large-scale expression studies included detergent extraction of FGFR2 from cell membranes. For every 100 µg of cell pellet or 100 µL of supernatant, 500 µL of lysis buffer with 1% detergent solution was added and resuspended in the presence of PMSF. The suspension for each was then constantly inverted for 2 h at 25 °C. The suspensions were then centrifuged at 20,800 g. Both pellet and supernatant were then stored at −80 °C. Several detergents were tested for optimal extraction from the cell pellet and the supernatant from the centrifugation: 1% n-dodecyl-β-D-maltopyranoside (DDM), 1% Brij 35, and 1% Brij 58 for the samples of pellet and supernatant. FGFR2/3 constructs were tested for binding to MBP-Trap HP affinity chromatography resin (GE Healthcare).

## Refolding by dialysis

Both FGFR2 and FGFR3 constructs containing ECD + TM were refolded by dialysis as described previously (*Mohammadi, Schlessinger & Hubbard, 1996*). The cell pellets were washed and resuspended with 0.5% guanidinium-HCl and centrifuged at 45,000 g for 20 min. Next, the pellets were solubilized in dialysis solution #1 (6 M guanidinium-HCl, 0.1% DDM, 10 mM DTT, and protease inhibitors E-64, benzamidine, PMSF, bestatin, and pepstatin at a pH of 8.0). To facilitate solubilization, the cell pellet and dialysis solution mixture was warmed briefly to 40 °C and then vortexed at room temperature. The total mixture was about 13 mL. All 13 mL of the solubilized inclusion bodies in the dialysis solution #1 was loaded into a dialysis membrane. This was placed in a beaker with 700 mL of dialysis solution #2 (25 mM HEPES, 150 mM NaCl, 10% glycerol, and 1 mM L-cysteine at pH 7.5) at 4 °C overnight with constant stirring using a magnetic stir bar. After 19 h, the sample within the dialysis membrane was then centrifuged at 24,000 g for 30 min and the supernatant was stored at −80 °C.

## FGF1 expression and purification

The FGF1 gene with an N-terminal His-tag in the expression vector pMCSG7 was obtained from the DNASU Plasmid Repository at Arizona State University. FGF1 was first purified using a 1 mL HiTrap GE Healthcare heparin affinity chromatography column, using elution buffer containing 1 M NaCl, 10% glycerol, 25 mM HEPES, 10 mM imidazole, and benzamidine with a pH of 7.5, as described previously (*Pellegrini et al., 2000*).

Successful purification by heparin affinity chromatography was confirmed by SDS-PAGE and Western blot analysis, but instead of using an anti-MBP antibody (HRP conjugated), an anti-His antibody (HRP conjugated) from Pierce was used. This was then followed by size exclusion chromatography on a Superdex 200 10/300 GL column (GE Healthcare). The running buffer used for size exclusion chromatography (SEC) was 25 mM HEPES, 0.1% DDM, and 150 mM NaCl.

### Initial functionality test of FGFR2

The first step in testing functionality is to determine whether FGFR2 can bind to heparin and FGF1. The 1 mL HiTrap Heparin Affinity Chromatography (GE Healthcare) column was used to test for heparin binding. To equilibrate the column, 10 column volumes (CVs) of binding buffer (150 mM NaCl, 25 mM HEPES pH 7.5, benzamidine, and 0.1% DDM) was loaded with a syringe. Next, 1 mL of FGF1 elution fraction from SEC was loaded onto the column and then 10 CVs of binding buffer. This was followed by FGFR2 supernatant from the dialysis. After loading, FGFR2 was washed with 5 CVs of binding buffer and eluted with 10 CVs of elution buffer (25 mM HEPES pH 7.5, 1.5 M NaCl, 0.1% DDM, benzamidine, and PMSF).

After chromatography, SDS-PAGE and western blot analysis were performed. The anti-MBP antibody was used to detect FGFR2, and the anti-His-tag antibody was used to detect FGF1. After heparin affinity chromatography, we performed size exclusion chromatography to assess the presence of aggregation, suggesting non-functional protein, or dimers, supporting functional protein.

## RESULTS

### Small-scale expression of FGFR2 and FGFR3 constructs

We performed small-scale expression trials of the FGFR2 and FGFR3 constructs in 100 mL of culture volume to determine which TM-containing construct would likely produce the highest yield for larger scale expression studies. Initial expression trials of FGFRs in the Rosetta2(DE3) strain of *E. coli* demonstrated extensive cell death after IPTG induction, suggesting toxicity of the expressed protein. We were able to express FGFRs and avoid expression toxicity using *E. coli* strain Lemo21, which contains T7 RNA polymerase that is titratable by rhamnose added to the media, a feature useful for expressing poorly folding membrane proteins and toxic proteins (*Wagner et al., 2008*; *Schlegel et al., 2012*). The western blot with an anti-MBP antibody showed significant quantities of FGFR2 and FGFR3 ECD + TM in both the soluble and cell pellet fractions (Fig. 2, lanes 2, 4, 7, 9). The intact receptors were not detected (data not shown). FGFR2 TM + KD was not detected (lanes 3, 8), but FGFR3 TM + KD (lane 10) was found in the cell pellet fraction. We considered the FGFR2 and FGFR3 ECD + TM constructs (lanes 7 and 9) to be the most promising for larger scale expression studies because of their superior yield, and the partial recovery of soluble FGFR3 ECD + TM (lane 4). In addition, for most of the constructs, prominent bands corresponding to the molecular weight of MBP were observed suggesting significant proteolysis of the fusion protein. This was not considered problematic as eventual structural studies would require removal of the MBP fusion-tag downstream of purification.

### Large-scale expression studies and detergent extraction analysis

After determining the optimal constructs from the small-scale expression studies, we performed expression trials of the FGFR2 and FGFR3 ECD + TM constructs in larger scale, 1L cultures of Lemo21 cells. We tested three detergent solutions containing 1% n-dodecyl-β-D-maltopyranoside (DDM), Brij 35, or Brij 58 for extraction of FGFR2/3
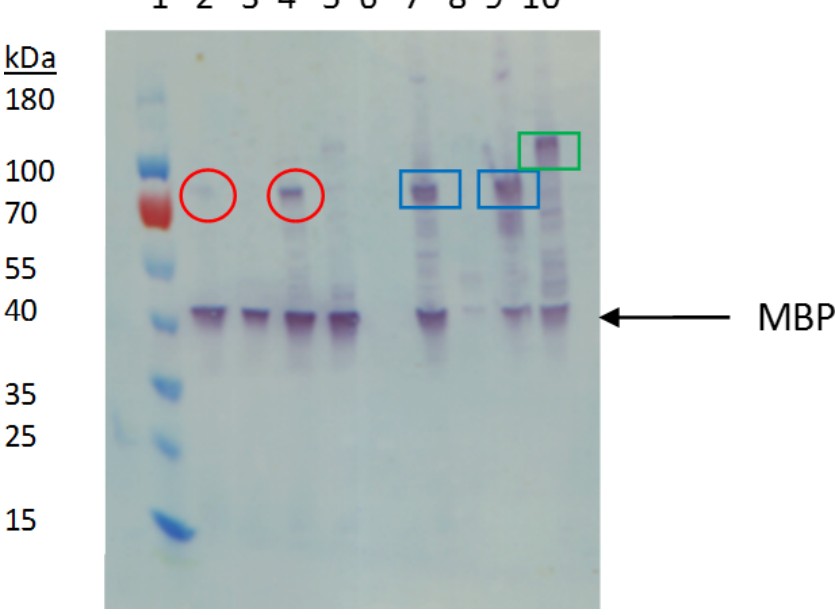

**Figure 2  Western blot analysis of small-scale expression of FGFR2 and FGFR3 constructs using anti-MBP antibody.** Lane 1: Ladder. Lane 2: FGFR2 31-406 from supernatant. Lane 3: FGFR2 370-651 from supernatant. Lane 4: FGFR3 143-405 from supernatant. Lane 5: FGFR 3: 365-771 from supernatant. Lane 6: Blank. Lane 7: FGFR2 31-406 from pellet. Lane 8: FGFR2 370-651 from pellet. Lane 9: FGFR3 143-405 from pellet. Lane 10: FGFR3 365-771 from pellet. Circled in red are bands consistent with FGFR2 and FGFR3 ECD + TM from supernatant. Boxed in blue are bands consistent with FGFR 2 and 3 ECD + TM from the cell pellet fraction. Boxed in green is a band consistent with FGFR3 TM + KD from the cell pellet fraction.

from cells. The western blot with an anti-MBP antibody on the detergent-extracted fractions showed significant quantities of FGFR2 and FGFR3 ECD + TM from both the soluble and cell pellet fractions (Fig. 3). We determined that DDM (lanes 2, 3, 8, and 9), Brij 35 (lanes 4, 5, 10, and 11), and Brij 58 (lanes 6, 7, 12, and 13) extracted FGFR2/3 to similar levels. We selected DDM for all subsequent procedures because it is the most commonly used detergent for membrane protein crystallography (*Privé, 2007*; *Loll, 2014*). As in the small-scale expression trials, we observed prominent bands corresponding to proteolyzed MBP in the western blots. Due to the large amounts of protein loaded, we also observed high amounts of non-specific binding in the western blot. We also observed a high molecular weight band that comigrated near the 180 kDa ladder band that we tentatively identify as oligomerized or aggregated FGFR2/3.

Based on the high expression levels shown on this western blot, especially from the soluble fraction, we initially decided FGFR3 ECD + TM would be our lead candidate for further expression and purification studies. However, we found that FGFR3 from the soluble fraction did not bind to the MBP affinity column. This suggested that the fusion protein, MBP-FGFR3 ECD + TM, was folded incorrectly, despite being soluble. Thus, we refocused efforts on recovering FGFR2 from the insoluble fractions. We pursued expression of FGFR2 ECD + TM in inclusion bodies and refolding by dialysis, as demonstrated previously for FGFR2 ECD (*Mohammadi, Schlessinger & Hubbard, 1996*). Refolding provided high yields

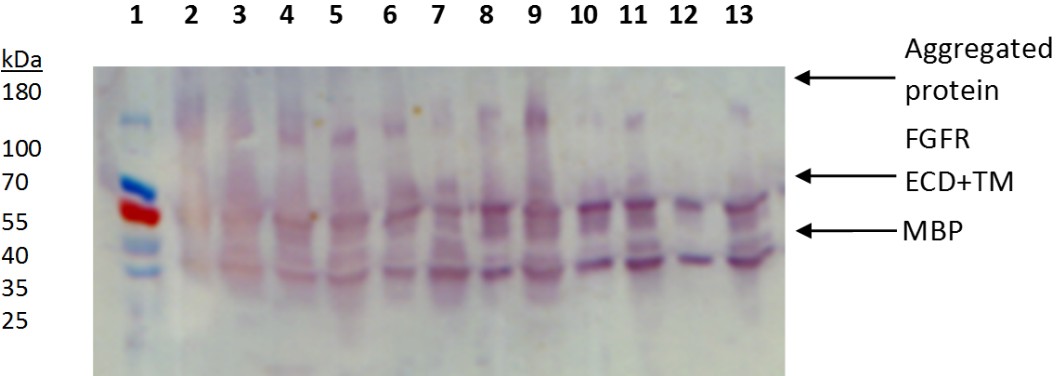

**Figure 3** Western blot of detergent extractions of large-scale expression constructs FGFR2 and FGFR3 ECD + TM. Lane 1: Ladder. Lane 2: FGFR2 pellet with 1% DDM. Lane 3: FGFR 2 supernatant with 1% DDM. Lane 4: FGFR2 pellet with 1% Brij 35. Lane 5: FGFR2 supernatant with 1% Brij 35. Lane 6: FGFR2 pellet with 1% Brij 58. Lane 7: FGFR2 supernatant with 1% Brij 58. Lane 8: FGFR3 pellet with 1% DDM. Lane 9: FGFR3 supernatant with 1% DDM. Lane 10: FGFR3 pellet with 1% Brij 35. Lane 11: FGFR3 supernatant with Brij 35. Lane 12: FGFR3 pellet with 1% Brij 58. Lane 13: FGFR3 supernatant with 1% Brij 58.

of FGFR2 ECD + TM, >4 mg of purified protein from 1 L of culture. The yield is adequate for protein crystallization.

## Binding of refolded FGFR2 to heparin

To test that the refolded FGFR2 ECD + TM retained its function, we sought to determine whether it could (1) bind heparin, (2) bind FGF1, and (3) dimerize. We tested the refolded fraction for binding to a heparin affinity chromatography column. A western blot with an anti-MBP antibody of the eluted fractions from the heparin affinity column demonstrated the presence of MBP-FGFR2 ECD + TM, supporting that the refolded FGFR2 bound heparin (Fig. 4).

## Refolded FGFR2 forms dimers but does not bind FGF1

The main elution fraction from the heparin affinity purification was then passed through a size exclusion chromatography column to resolve its components (Fig. 5). The first peak, eluting at 25.76 mL, corresponds to a molecular weight of 200 kDa. The second peak, eluting at 30.79 mL, corresponds to between 66 and 79 kDa. The third primary peak, eluting at 34.99 mL, corresponds to a size between 12 and 20 kDa. Each of these three primary peaks was analyzed by SDS-PAGE and western blots (Fig. 6). The second peak corresponded to the molecular weight of the MBP-FGFR2 ECD + TM construct (71.5 kDa) and was identified by western blot with an anti-MBP antibody (data not shown). The third peak corresponded to FGF1 from its molecular weight (17.5 kDa) and was identified by western blot with an anti-His-tag antibody (data not shown).

We considered two possibilities for the identity of the first peak: (1) a complex of MBP-FGFR2 dimer with FGF1, (2) a dimer of MBP-FGFR2 in DDM micelles. Western blot analysis with an anti-MBP antibody confirmed the presence of MBP-FGFR2 ECD + TM (Fig. 6). We eliminated the possibility of the peak being the FGFR2-FGF1 complex because the expected molecular weight is 230.5 kDa, and western blot analysis with an

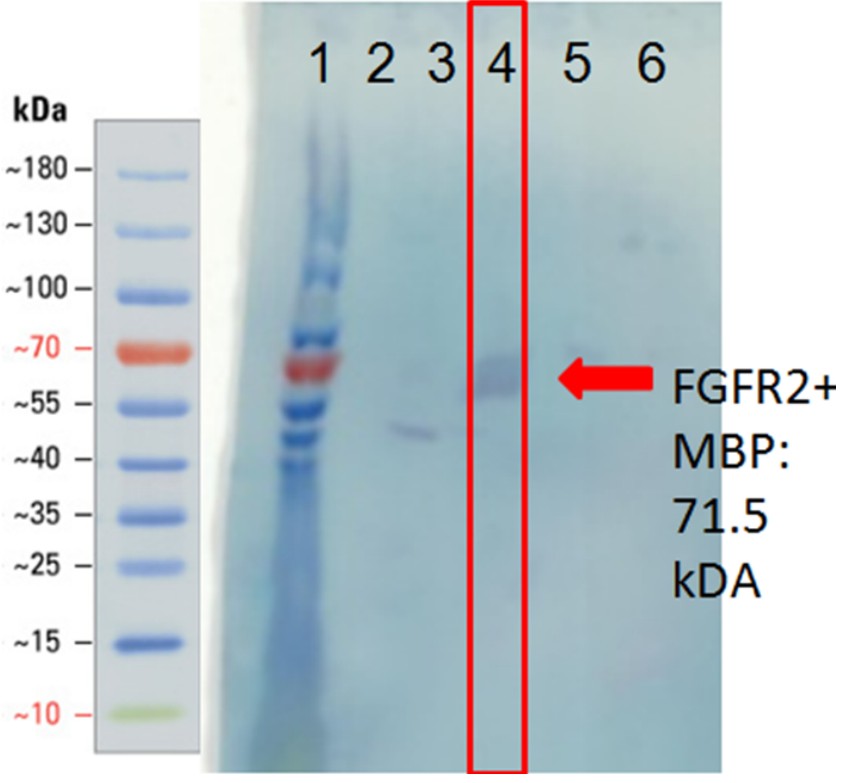

**Figure 4** **Western blot of heparin affinity column purification fractions using anti-MBP antibody.** Lane 1: ladder. Lanes 2–3: wash fractions. Lanes 4–6: elution fractions.

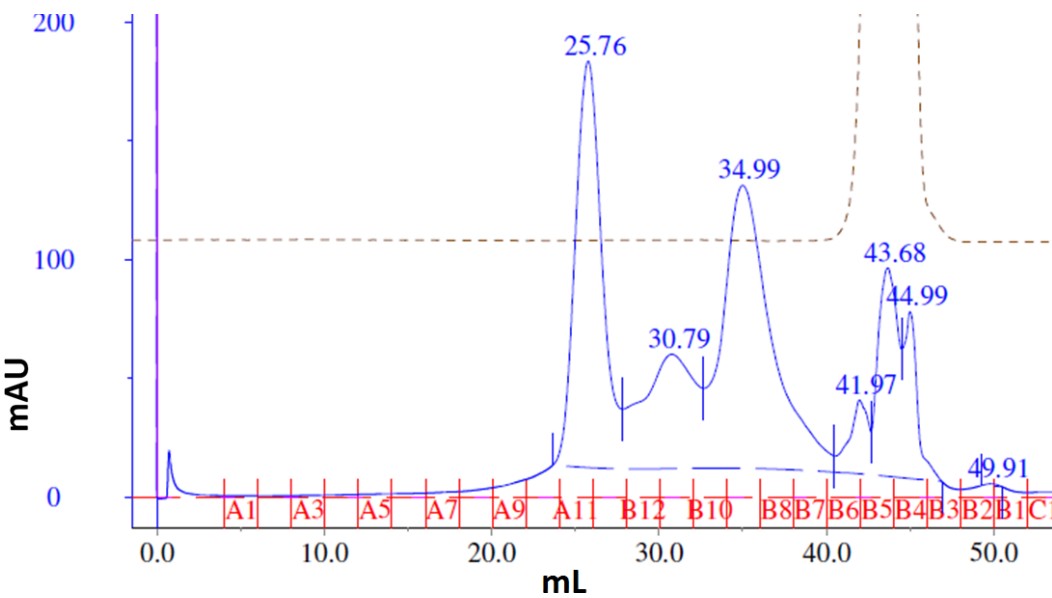

**Figure 5** **Size exclusion chromatography of the main heparin affinity elution fraction.** Elution fractions are marked in red.
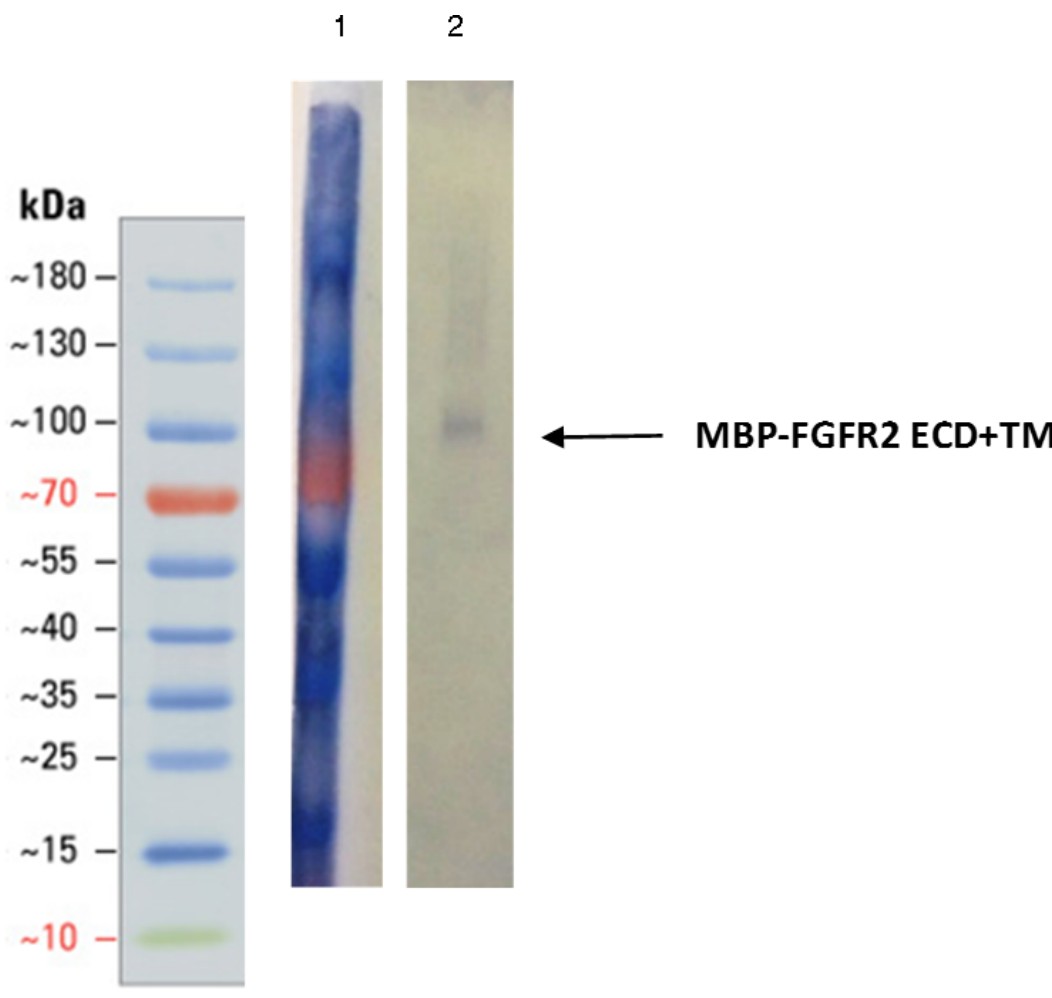

**Figure 6 Western blot of the first peak from SEC using anti-MBP antibody.** Arrow points to band correlating to FGFR2 construct size (71.5 kDa). Lane 1: Ladder. Lane 2: SEC fraction A12. Both lanes are cropped from the same blot.

anti-His-tag antibody did not show the presence of FGF1 (data not shown). In contrast, the expected molecular weight of an MBP-FGFR ECD-TM dimer with a DDM micelle is 213 kDa. Potentially, the inclusion of the TM region or DDM may stabilize an inactive conformation of the dimer, rendering it incapable of binding FGF1. Another possibility is that the MBP fusion-tag interfered with FGF1 binding.

## CONCLUSIONS

Our results present progress toward recombinantly expressing partially functional FGFR2 ECD + TM in *E. coli*. This is the first report of recombinant expression in *E. coli* of a eukaryotic protein kinase construct containing its TM domain. Protein production in *E. coli* is highly desirable because of low costs, fast growth, easy mutagenesis, and high protein yields. Key steps include the use of the MBP fusion tag, use of the Lemo21 (DE3) strain, refolding from inclusion bodies, and use of the detergent DDM throughout all extraction

and purification procedures. The purified FGFR2 ECD + TM demonstrated the ability to dimerize and bind heparin but did not form a stable complex with FGF1 as observed by size exclusion chromatography. This may suggest that the purified FGFR2 was not fully folded or functional. Other possible explanations include: (1) inclusion of the TM or detergent favors an inactive conformation, (2) stable complex formation requires the addition of accessory molecules such as heparin, heparan sulfate, or sodium octasulfate (*Zhang et al., 2009*), or (3) the MBP fusion tag interfered with FGF1 binding. We plan on future experiments to address these possibilities. The potential inhibitory role of the TM in FGF binding and receptor activation merits further investigation. Future studies of FGFR and receptor kinase function should include the TMs in the expressed protein constructs as its biochemical role is increasingly recognized.

### Funding

This project was funded by NSF CAREER Award 1350555 (HLN), the University of Hawaii at Manoa, and the University of Hawaii at Manoa Undergraduate Research Opportunities Program (AB). The funders had no role in study design, data collection and analysis, decision to publish, or preparation of the manuscript.

### Grant Disclosures

The following grant information was disclosed by the authors:
NSF CAREER: 1350555.
University of Hawaii at Manoa.
University of Hawaii at Manoa Undergraduate Research Opportunities Program (AB).

### Competing Interests

The authors declare there are no competing interests.

### Author Contributions

- Adam Bajinting conceived and designed the experiments, performed the experiments, analyzed the data, wrote the paper, prepared figures and/or tables, reviewed drafts of the paper.
- Ho Leung Ng conceived and designed the experiments, analyzed the data, wrote the paper, prepared figures and/or tables, reviewed drafts of the paper.

### Data Availability

The raw data has been supplied as images of electrophoretic gels and blots.

### Supplemental Information

Supplemental information for this article can be found online at http://dx.doi.org/10.7717/peerj.3512#supplemental-information.

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
