# Peer review of "Recombinant expression in E. coli of human FGFR2 with its transmembrane and extracellular domains"

_PeerJ, doi:10.7717/peerj.3512_

## Round 0.1 · original submission · Major Revisions

please analyse and discuss the problems during FGFR proteins expressed in E. coli

·

Basic reporting

Clear and unambiguous, professional English is used.
p30 line 20 ff has the verb follow in 3 subsequent sentences.

11 references are provided which is sufficient to introduce the biological background and context of FGFR Proteins. The topic of the paper, namely how to overcome the problem of recombinant expression and subsequent purification is not sufficiently represented (1 citation).

The article structure, figs and tables are in accordance with the requirements of the subject.

The research hypothesis that FGFR Proteins can be expressed recombinantly in E. coli was verfied. The problems are described but not analyzed.

Experimental design

The expression and purification of mammalian and membrane proteins is original, primary research within the aims and scope of the journal. It is stated that the presented research is required to overcome the knowledge gap with respect to structual biology, but not with respect to the actual work: The observed fragmentation of the expression construct was not analyzed with regard to the FGFRs. Specifically, a mass spectrometry of the isolated product or an other acessment of the completeness of the isolated product is missing.
Some further details with regard to replication are missing:
p17 line 15: type of SEC column
p10 line 12: the rotor type or g-force instead of rpm

The investigation was performed with standard methods that are well established.

Validity of the findings

The observed fragmentation was not analysed with regard to is relevance for the product.
The integrity of the product is assumed based on its capability to bind heparin.
The conclusions are well stated, link to the research question and the results.

Additional comments

The problem of fragmentation to the identity of the isolated product is serious, especially if structural analysis is intended. It is not clear if the protein in Fig 4 shows as a double band.

·

Basic reporting

Writing quality:
The manuscript English is generally well written and clear.
Italicize E. coli and in vitro throughout the text.

Figure & Table Quality:
A comment in Table 1 on providing the reference number from bp=0 as it relates to the base pair ranges in the constructs would be helpful.

A reference to figure 2 lanes in the text would be particularly appropriate for the “most promising” constructs (P10,L12). Normally I would recommend colored boxes on gel by comparison to side-bar notations.

Raw Data:
An explicit reporting of protein expression and subsequent yield estimates in purification would be a positive addition to the manuscript.

Experimental design

Journal Scope:
The manuscript is in the “Biological Sciences” scope of the journal.

Relevance:
The use of Lemo21 line and eventual focus on inclusion body refolding is an inconsistency that should be discussed as it influences interpretation of the significance of results (focused on making progress towards functional expression). A clarification on use of Lemo21 cells which is designed to avoid inclusion body formation though rhamnose modulated expression of T7 RNA polymerase inhibitor (Lysy). In this context, the level of rhamnose should be specified as it is not a standard ingredient of the TB (Terrific Broth) media used in this work.

Rigorous execution:
Considerable effort is made starting with a variety of expression constructs. This approach then focusing down on the best candidates is a strong approach; though there is little to no discussion on what was learned from this approach.

Methods Description:
Methods related to cell lysis are presented in three different methods sections. Clarify small scale, large scale and if the detergent extraction procedure is independent or supplemental to the preceding lysis procedure (p.7).

Details of sonification intensity, configuration, equipment would be useful to reproduce results.

The final methods description for Functional testing ends with in inadequately rationalized description (P9/10). Two sentences in a row “This was followed”, with the final sentence not providing rationale for why SEC is used. An ending such as “to assess bound / unbound FGF ligand, dimers … ?” would provide a much better transition to results.

Validity of the findings

Conclusions appropriate:

Suggest improving first sentence of conclusions to read: “Our results present progress toward expression partially functional”

The final sentence of the conclusions is a weak / ambiguous ending: “The potential role of the TM merits further investigation”. Is this a suggestion to remove the transmembrane domain to study binding characteristics.

Give the moderate success with considerable effort at expressing this protein in E. coli, it would seem to be worth noting (either as conclusion or background) the potential value of pursuing alternative expression platforms that are more specifically developed for trans-membrane protein expression (e.g. Rhodobacter www.ncbi.nlm.nih.gov/pubmed/26008117, or cell-free systems link.springer.com/protocol/10.1007/978-1-60761-344-2_11

Speculation identified: The role of TM is implicitly speculated in conclusions, and should be addressed in the context of comment above.

Additional comments

As other work has been done with expression with this protein, it would seem appropriate to provide some reference to that work in the DISCUSSION to build upon the background in which other work is noted.

Suggested edits:
P4,L8: change “The strongest data supporting or invalidationg” to “Resolving different”
P.6,L5: space before units “mM”.
P9,L8: Change “Determination of success was” to “Successful”; suggest sentence structure correction to semi-colon separating the to method and clarification “Western blot analysis; instead of using ...”; insert verb to read “was confirmed by SDS-Page”.
P9,L17: Move the ending preposition “to equilibrate column” to replace the non-description start of the sentence “First,” ; delete the extra “onto the column”.
P11,L13: replace “following” with “subsequent”
P14,L18: replace “not capable” with “rendering it incapable”

---

## Round 0.2 · accepted · Accept

I recommend "Accept" for the revised manuscript entitled "Recombinant expression in E. coli of human FGFR2 with its transmembrane and extracellular domains"

·

Basic reporting

no comment

Experimental design

no comment

Validity of the findings

no comment